# Acidity Suppression of Hole Transport Layer via Solution Reaction of Neutral PEDOT:PSS for Stable Perovskite Photovoltaics

**DOI:** 10.3390/polym12010129

**Published:** 2020-01-06

**Authors:** Minseong Kim, Minji Yi, Woongsik Jang, Jung Kyu Kim, Dong Hwan Wang

**Affiliations:** 1School of Integrative Engineering, Chung-Ang University, Seoul 06974, Korea; kms4587@cau.ac.kr (M.K.); yi4110@cau.ac.kr (M.Y.); dndtlr2@cau.ac.kr (W.J.); 2School of Chemical Engineering, Sungkyunkwan University (SKKU), Suwon-si 16419, Korea

**Keywords:** conducting polymers, photovoltaic devices, controlled pH, charge transport, stability

## Abstract

Poly(3,4-ethylenedioxythiophene): poly(4-styrenesulfonate) (PEDOT:PSS) is typically used for hole transport layers (HTLs), as it exhibits attractive mechanical, electrical properties, and easy processability. However, the intrinsically acidic property can degrade the crystallinity of perovskites, limiting the stability and efficiency of perovskite solar cells (PSCs). In this study, inverted CH_3_NH_3_PbI_3_ photovoltaic cells were fabricated with acidity suppressed HTL. We adjusted PEDOT:PSS via a solution reaction of acidic and neutral PEDOT:PSS. And we compared the various pH-controlled HTLs for PSCs devices. The smoothness of the pH-controlled PEDOT:PSS layer was similar to that of acidic PEDOT:PSS-based devices. These layers induced favorable crystallinity of perovskite compared with acidic PEDOT:PSS layers. Furthermore, the enhanced stability of pH optimized PEDOT:PSS-based devices, including the prevention of degradation by a strong acid, allowed the device to retain its power conversion efficiency (PCE) value by maintaining 80% of PCE for approximately 150 h. As a result, the pH-controlled HTL layer fabricated through the solution reaction maintained the surface morphology of the perovskite layer and contributed to the stable operation of PSCs.

## 1. Introduction

The advances in organic optoelectronics rely on the ability to control various transport properties of devices by controlling the morphology and interface deformation and changing charge carrier density and mobility. Device performance is related to the electrical properties and morphologies of the organic materials used to fabricate the devices [1]. Furthermore, the ability to fabricate ultra-thin films with thicknesses in the nanometer scale and with smooth surfaces is advantageous for the fabrication of laminated structures and applications [2,3,4,5,6]. Among the conducting polymers, the most promising material is poly(3,4-ethylene dioxythiophene) (PEDOT) doped with poly(styrene sulfonate) (PSS) (PEDOT:PSS), which consists of a conducting polythiophene derivative bound to a PSS polyanion [7,8]. Because this polymeric system simultaneously has high electrical conductivity, high mechanical flexibility, and good transparency in the visible range [7,8,9,10,11,12,13], it has been widely applied for organic photovoltaics, flexible electrodes, light-emitting diodes, field-effect transistors, and thermoelectric generator [7,14,15,16,17,18,19,20,21,22].

PEDOT:PSS is intrinsically an acidic and hygroscopic material because of its unique chemical structure and hydrophilicity. The high acidity of PEDOT:PSS, which is the most promising hole transport layer (HTL) material in optoelectronic devices, can corrode indium tin oxide (ITO) electrodes, degrade the perovskite, and deteriorate device performance and stability [9,10,11,13,23,24,25,26,27]. Previous approaches to solve this problem include neutralizing acidic PEDOT:PSS by strong bases such as NaOH and KOH or using imidazole as an additive [28,29,30,31]. The fabrication of PEDOT:PSS depends on various factors, including the ratio of PSS, concentration of solid contents, colloid gel particle size, and viscosity, which can affect the electronic properties and morphologies in the film. In this context, studying the correlation between morphology, solid additives, and fraction of PSS can solve the problems associated with device application.

The residual acidic agents and proton in acidic PEDOT:PSS causes the problem of deterioration of ITO or top electrode and degradation of the perovskite which leads to a decrease of perovskite crystal quality [27,32]. The high viscosity of neutral PEDOT:PSS leads to unsuitable surface morphology for optoelectronic devices. We propose a convenient and successful approach to suppress the acidic property of PEDOT:PSS, using a simple solution reaction that controls the volume ratio of two PEDOT:PSS solutions, acidic, and neutral PEDOT:PSS. We evaluated the effect of pH of PEDOT:PSS on key features that affect the size of colloidal gel particles (tertiary structure) [33], the morphology of PEDOT:PSS, the perovskite active layer, and electrical properties of perovskite solar cells (PSCs). Our results suggest that the use of PEDOT:PSS using the optimal volume ratio of acidic and neutral PEDOT:PSS can not only improve device stability but can also enhance the polymerization degree of PEDOT and increase perovskite crystallinity.

## 2. Materials and Methods

### 2.1. Material Preparation

Methylammonium iodide (CH_3_NH_3_I, Dyesol-Timo Co. Ltd., Seongnam, South Korea), lead iodide (PbI_2_, 99%, Sigma Aldrich, St. Louis, MO, USA), γ-butyrolactone (GBL, Sigma Aldrich, St. Louis, MO, USA), and dimethyl sulfoxide (DMSO, Sigma Aldrich, St. Louis, MO, USA) were used to prepare the CH_3_NH_3_PbI_3_ (MAPbI_3_) precursor. Acidic PEDOT:PSS (AI 4083, Heraeus Company, Hanau, Germany) and neutral PEDOT:PSS (Neutral PEDOT:PSS, Sigma Aldrich, St. Louis, MO, USA) were used as HTLs. Imidazole present in the neutral PEDOT:PSS affects the pH of the HTL solution. PCBM (6,6-phenyl-C70 butyric acid methyl ester)/titanium(VI) isopropoxide (TiO_x_, Sigma Aldrich, St. Louis, MO, USA) was used as the electron transport layer in PSCs.

### 2.2. Thin Film and Device Fabrication

The PSCs were prepared on ITO substrates, which were cleaned using detergent and underwent sonication in deionized water, acetone, and isopropanol. After exposure of ITO substrates to UV ozone for surface modification, the PEDOT:PSS was spin-coated on the ITO substrates and annealed to fabricate a 30-nm-thick thin film. CH_3_NH_3_I and PbI_2_ (1.06:1 mol %) were dissolved in DMSO: GBL (3:7 *v*/*v*, molar concentration of 1.4 mol L^−1^). The MAPbI_3_ solution was spin-coated on the PEDOT:PSS thin film, followed by additional treatment with chlorobenzene. Then, the substrates were placed on a hot plate at 100 °C to fabricate a MAPbI_3_ film with a thickness of 300 nm. The PCBM solution was dissolved in chlorobenzene (20 mg mL^−1^) and spin-coated onto the surface of MAPbI_3_ at a thickness of 100 nm. To fabricate a 30-nm-thick PCBM layer, the PCBM solution was dissolved in chlorobenzene (20 mg mL^−1^) and spin-coated on the MAPbI_3_ layer. Subsequently, a TiO_X_ layer (~10 nm) was deposited. Finally, an Ag cathode was deposited by thermal evaporation.

### 2.3. Device Characterization 

The pH values of PEDOT:PSS were measured using a pH meter (HANNA instruments, Inc., HI8424, Seoul, South Korea). The zeta radii of different ratios of acidic and neutral PEDOT:PSS were examined using a Malvern Zetasizer ZS90 (ZEN3690) to investigate the particle sizes of PEDOT:PSS in the water of each sample. Fourier transform infrared (FTIR) spectra, collected using horizontal attenuated total reflectance (HATR), were recorded on a NICOLET 6700 IR spectrometer. The electrical properties of the PSCs were measured under air mass 1.5 global illumination (AM 1.5 G) using a solar simulator (Peccell Technologies, PEC-L01) at an intensity of 100 mW cm^−2^, which was calibrated using a silicon reference cell. The current density–voltage characteristics of the PSCs were measured using ZIVE SP1. After power calibration (ABET Technologies, Inc., LS150, Milford, CT, USA) using a monochromator (Dongwoo Optron Co. Ltd., MonoRa-500i, Gwangju, South Korea), the EQE was measured. The surface morphology and roughness of spin-coated films were examined using an atomic force microscope (AFM) in the noncontact mode (Park NX10) and a scanning electron microscope (SEM) (SIGMA model from Carl Zeiss, Inc., Oberkochen, Germany).

## 3. Results and Discussion

The PEDOT:PSS has the chemical structure and general arrangement [34,35,36] as shown in Figure 1a. The PEDOT:PSS consists of two ionic bond polymers presented in chemical structure. And when the PEDOT:PSS is deposited as a thin film, as the PEDOT is bonded to PSS chains via Coulomb interactions, π-stacking structure between PEDOT and PSS is induced, which improves hole transport property and conductivity [37,38]. The PEDOT:PSS exhibits strong acidic properties and coating a solution with strong acidic PEDOT:PSS can deteriorate the ITO by penetrating the boundary between the solution and the thin film to promote ion release and impede charge flow [11,27,39,40,41]. Further, the acidic property of PEDOT:PSS can cause the degradation of the top electrode or perovskite active layer and lead to inferior perovskite crystal quality [9,10,11,13,24,25]. In this study, we minimized the strong acidic properties of PEDOT:PSS and improved the stability of PSCs by controlling the pH of the PEDOT:PSS solution. By mixing acidic PEDOT:PSS and neutral PEDOT:PSS in an optimal ratio, we verified the efficacy of the PEDOT:PSS solution that can be easily prepared. The pH of acidic PEDOT:PSS is approximately 1–2 and the viscosity is approximately 5–12 mPa s. In addition, the pH of neutral PEDOT:PSS is approximately 5–7, and the viscosity is less than 100 mPa s. As depicted schematically in Figure 1b, pH-controlled PEDOT:PSS solutions were prepared by mixing acidic and neutral PEDOT:PSS in the volume ratios of 1:0 (AN10), 1:1 (AN11), and 1:3 (AN13). By mixing acidic and neutral PEDOT:PSS, the pH of the solution increased, and the resistivity decreased, thus enhancing the stability and efficiency of PSCs.

The PEDOT:PSS consists of gel-like particles containing a PSS^-^ shell which stabilizes PEDOT-rich particles in an aqueous solvent [42]. The size of the colloidal PEDOT:PSS gel particle in water is related to the ratio of PEDOT to PSS, as increasing particle size increases the particle boundary surface [33,35,43], see Figure 1b. An increase in the particle size with decreased particle boundaries leads to a few energy barriers which enhance the conductivity of PEDOT:PSS [44,45]. Three types of particles of the PEDOT:PSS colloidal gel—AN10, AN11, and AN13—were examined by measuring the zeta-average particle size of each sample (Figure 2). The average particle size of AN11 and AN13 PEDOT:PSS was found to be 326.9 and 382.0 nm, respectively, whereas the average particle size of AN10 PEDOT:PSS was found to be 245.6 nm. As can be seen from the size distribution of PEDOT:PSS, the particle size increased by mixing acidic and neutral PEDOT:PSS (depicted in Figure 1b). The particle size of PEDOT:PSS in water highly depends on the ratio of PEDOT to PSS. These results illustrate that neutral PEDOT:PSS has a higher PEDOT ratio in comparison to acidic PEDOT:PSS. In addition, neutral PEDOT:PSS and acidic PEDOT:PSS were found to have mixed well.

As the gel particle polymer composite in aqueous solution is deposited and annealed forming thin PEDOT:PSS film [43,46,47,48,49], the PEDOT chains interact with the neighboring PEDOT-rich domains and bring the conducting domains closer by coalescence of PEDOT:PSS particles. The thermal annealing evaporates and softens PSS-rich domains, which improve connectivity and charge transport of conducting domains [47,50,51]. After the annealing and additional polymerization, the phase separation of PEDOT and PSS occurs. Furthermore, it increases the continuous π-stacking of PEDOT:PSS and provides a pathway where charge carrier can flow [37,38,52]. To identify the degree of polymerization of PEDOT and the different vibrational modes of various bonds, which are present in PEDOT:PSS layer, the spin-coated films on a silicon wafer are studied using FTIR spectroscopy (Figure 3). The FTIR spectra showed the existence of functional bonds of PEDOT:PSS after annealing at 140 °C. The typical bands of PEDOT:PSS were observed at 1277, 1173, 1133, 1003, 966, 920, 825, and 685 cm^−1^. C–S–C bond vibrations were observed to occur at 966, 920, 825, and 685 cm^−1^. The bands at 1190 and 1003 cm^−1^ were attributed to the S=O and O–S–O symmetric stretching modes in PSS, respectively. As the ratio of the neutral PEDOT:PSS increased, the peak integration at 1277 cm^−1^ increased caused by the imidazole content in neutral PEDOT:PSS. Appendix A shows the peak integration and ratio of each typical band. In addition, these bands were clearly observed, which denotes that the spin-coating and annealing processes for device fabrication was carried out without degradation of the polymer. 

To compare the degree of electrochemical polymerization occurs at the α,α’-positions in polythiophene [53], the polymerization degree of PEDOT was evaluated from the ratio of integration of the infrared bands [54]. The average degree of polymerization of PEDOT can be evaluated using the equation as follows [55,56,57],
$$\mathrm{Degree}\;\mathrm{of}\;\mathrm{polymerization}=2\left(\frac{R_{0}}{R}+2\right)$$
where R is the integrated intensity ratio of the IR bands at 685 and 825 cm^−1^, which indicates the characteristic bands of stretching vibrations of the C-S-C bond and R0 is 1.07, which is determined for α-sexithiophene. The films were ordered by their degree of polymerization as follows: AN11 PEDOT:PSS < AN10 PEDOT:PSS < AN13 PEDOT:PSS, indicating AN11 PEDOT:PSS exhibited the highest degree of polymerization. Thus, AN11 was found to be an efficient interlayer for PSCs with optimized PEDOT and PSS ratios and controlled pH values. For the fabrication of practicable and stable PSCs, an appropriate amount of imidazole and a high degree of polymerization is necessary. Our results indicate that the varied properties of AN10, AN11, and AN13 induce the different degree of polymerization which affects the carrier mobility of PEDOT:PSS [54,58,59]. Based on the above result, by mixing an optimized amount of neutral and acidic PEDOT:PSS, the AN11 PEDOT:PSS is expected to exhibit enhanced charge carrier mobility because of improved conjugation and polymerization with π–π stacking structure which improves the electrical properties of PSCs.

The properties of mixed PEDOT:PSS such as viscosity, pH value, and colloidal gel particle size can affect the thickness and roughness of the spin-coated layer. The thickness of spin-coated PEDOT:PSS films were found to be 31–37 nm for AN10, AN11, and AN13 using the Dektak XT thickness profiler (Appendix A). The absorbances of the films were compared (Appendix A), and those of AN11 and AN13 were similar in the visible region compared to films coated with AN10. We observed that the different properties of AN11 and AN13 did not affect the thickness and absorbance of the films. In addition, AFM topography images were obtained for different PEDOT:PSS spin-coated films (AN10, AN11, and AN13) to allow studying the possible changes in morphology at the nanometer scale (Figure 4). The root-mean-square (RMS) roughness values of AN10, AN11, and AN13 were found to be 0.28 nm, 0.32, and 0.34 nm, respectively. Furthermore, the AFM images showed a uniform and reasonably smooth surface of PEDOT:PSS film. Therefore, when different PEDOT:PSS solutions with various viscosity were mixed in a certain ratio, the thickness and surface roughness of the AN11 and AN13 films were compatible with those of the AN10 film.

Furthermore, perovskite crystals were spin-coated and analyzed for each HTL prepared by varying the volume ratios of the acidic and neutral solutions. The perovskite thin film prepared on the PEDOT:PSS layer was found to be flat, pin-hole-free, and covered the entire substrate. The perovskite film formed on the acidic PEDOT:PSS layer displayed a crystalline morphology with a perovskite crystal grain size of approximately 84 nm. As shown in Figure 5, the grain size of the AN11/perovskite film was approximately 113 nm; this result confirmed the formation of a larger grain size of the perovskite film. However, in the AN13/perovskite films, grain sizes larger than 200 nm were rarely observed, and the grain size appeared to have reduced (average 96 nm). In perovskite crystals, grain boundaries are known to provide trap sites for charges, which is disadvantageous in terms of charge transfer [60,61,62]. Therefore, we confirmed that AN11 samples were optimized for the smooth operation of perovskite photovoltaic cells and could also be advantageously used for perovskite crystal formation compared to AN10 and AN13.

The crystallinity of the perovskite layer is a decisive factor for device performance. To reveal the crystallinity of perovskite deposited on HTLs, X-ray diffraction (XRD) was performed, and the results are shown in Figure 6. The XRD patterns of the glass/PEDOT:PSS/MAPbI_3_ films showed identical diffraction peaks at 14.0° (110), 28.3° (220), and 31.7° (310), which are characteristic peaks of tetragonal MAPbI_3_ perovskite crystals. Thus, PbI_2_ was found to completely react with CH_3_NH_3_I to form a pure perovskite phase. The full width at half maximum (FWHM) is inversely proportional to the crystal grain size, according to the Scherrer equation. The diffraction peak at 13.99° shows that the trend of FWHM is correlated with the grain size distribution of the perovskite layer in Figure 5. In addition, Figure 6a shows that the intensity of the PbI_2_ diffraction peaks at 12.7° increased in acidic PEDOT:PSS [63,64,65], indicating the formation of the PbI_2_ phase because of the acidity of PEDOT:PSS leading to degradation of perovskite. Note that the pH-controlled PEDOT:PSS induces higher perovskite crystallinity and less degradation of the perovskite layer.

The band energy alignment and schematic structure of the PSC devices (Device structure: ITO/PEDOT:PSS/MAPbI_3_/PCBM/TiO_X_/Ag) are depicted in Figure 7a,b. Figure 7c,d shows the current-voltage (J–V) curve and the external quantum efficiency (EQE) of the PSCs manufactured using pH-modified HTLs by mixing acidic and neutral PEDOT:PSS. Table 1 shows the electrical parameters of devices based on Figure 7c,d. For PSCs fabricated with the AN10 PEDOT:PSS layer, the open-circuit voltage (V_OC_), short-circuit current (J_SC_), fill factor (FF), and power conversion efficiency (PCE) were 0.957 V, 15.28 mA cm^−2^, 70%, and 10.31%, respectively. Meanwhile, PSCs fabricated with the AN11 PEDOT:PSS layer exhibited increased V_OC_, FF, and PCE values: 0.975 V, 71% and 10.34%, respectively. However, the J_SC_ value of PSCs fabricated with the AN11 PEDOT:PSS layer was similar to that of PSCs fabricated with the AN10 PEDOT:PSS layer (14.98 mA cm^−2^). Furthermore, the PSCs fabricated with the AN13 PEDOT:PSS layer showed an increased V_OC_ of 1.006 V but exhibited decreased J_SC_, FF, and PCE: 13.94 mA cm^−2^, 68%, and 9.60%, respectively. The EQE data in Figure 7d confirm the similar J_SC_ values of the PSCs, which were 14.56, 13.69, and 12.59 mA cm^−2^, respectively, for the PSCs fabricated with AN10, AN11, and AN13 PEDOT:PSS layers. As the ratio of neutral PEDOT:PSS increased, the V_OC_, measured by cyclic voltammetry, slightly increased because of the modulated work function of the PEDOT:PSS layer by mixing acidic and neutral PEDOT:PSS solutions (Appendix A). The significant decrease in the J_SC_ value with an increase of neural PEDOT:PSS ratio can be explained by the low crystallinity of perovskite. Also, the decrease of FF and PCE in neutral PEDOT:PSS-based PSCs can be explained by low crystallinity and unsuitable morphology induced by high viscosity (Appendix A). PSCs with AN13 PEDOT:PSS HTLs have low J_SC_ value. In addition, FF was slightly improved in the AN11 PEDOT:PSS-based device and was the lowest in the AN13 PEDOT:PSS-based device, which can be correlated with the crystallinity and grain size of the perovskite active layer. Therefore, for the smooth and stable operation of perovskite photovoltaic cells, the optimal ratio of acidic and neutral PEDOT:PSS solutions needs to be identified. We propose an optimal ratio of acidic to neutral PEDOT:PSS solutions of 1:1, obtained by manufacturing a device that could maintain the PCE equivalent to that of acidic PEDOT:PSS.

Finally, stability analysis was performed to confirm the effect of improved quality and stability of PSCs. As shown in Figure 8, the effect was analyzed in a device stability test for approximately 200 h. The AN10- and AN13-based devices maintained ~80% efficiency after 50 h. However, after 50 h, the efficiency of the AN10-based devices dropped sharply, and that of the AN13-based device dropped rapidly within approximately 70 h. The increased ratio of neutral PEDOT:PSS may cause decreased stability in AN13-based PSCs because of low crystallinity and unsuitable morphology of MAPbI_3_ perovskite. In contrast, AN11-based devices maintained 80% efficiency over 150 h. We also maintained the PCE level at 60% when the AN10- and AN13-based devices were not working. Thus, a slight increase in the PCEs compared with those of the acidic PEDOT:PSS-based device was found to be due to the improved V_OC_ and FF and similar J_SC_. Furthermore, the optimized pH-controlled PEDOT:PSS was superior to acidic PESOT: PSS in terms of long-term stability of PSCs because pH-controlled PEDOT:PSS suppresses the release of indium ions in the ITO electrode and inhibits degradation of the perovskite.

## 4. Conclusions

We suppressed the acidity of PEDOT:PSS by blending acidic and neutral PEDOT:PSS solutions at the optimal volume ratio which enhanced the stability and efficiency of PSCs. By mixing acidic and neutral PEDOT:PSS solutions, the particle size of the PEDOT:PSS colloidal gel changed, which was directly related to the ratio of PEDOT and PSS. As the ratio of neutral PEDOT:PSS increased, the acidic properties of PEDOT:PSS decreased, which enhanced the degree of polymerization. With the pH-controlled PEDOT:PSS, PSCs showed improved V_OC_ values and similar levels of J_SC_ compared with those of acidic PEDOT:PSS-based PSCs. This was achieved by better control of the PEDOT:PSS layer and the crystallization process of MAPbI_3_ through the successful change in properties of pH-controlled HTLs. In addition, AN11 PEDOT:PSS-based devices retained 80% of the PCE in the 150-h stability test because of the improved morphological properties of the MAPbI_3_, whereas the PCE of PSCs fabricated with AN10 PEDOT:PSS layers fell to 20% of its initial value. Our approach for producing pH-controlled PEDOT:PSS provides a clear direction for research by providing a method to improve the stability of PSCs and can be a useful approach for solving strong acid-related problems in various conductive polymers.

## Figures and Tables

**Figure 1 polymers-12-00129-f001:**
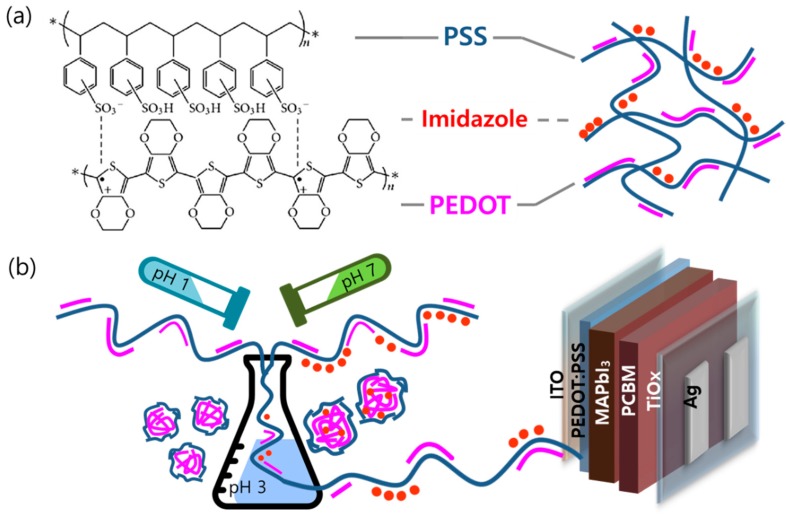
(**a**) Structure and schematic representation of PEDOT:PSS. (**b**) The solution fabrication process and polymer composite structure of pH-controlled PEDOT:PSS for inverted PSCs.

**Figure 2 polymers-12-00129-f002:**
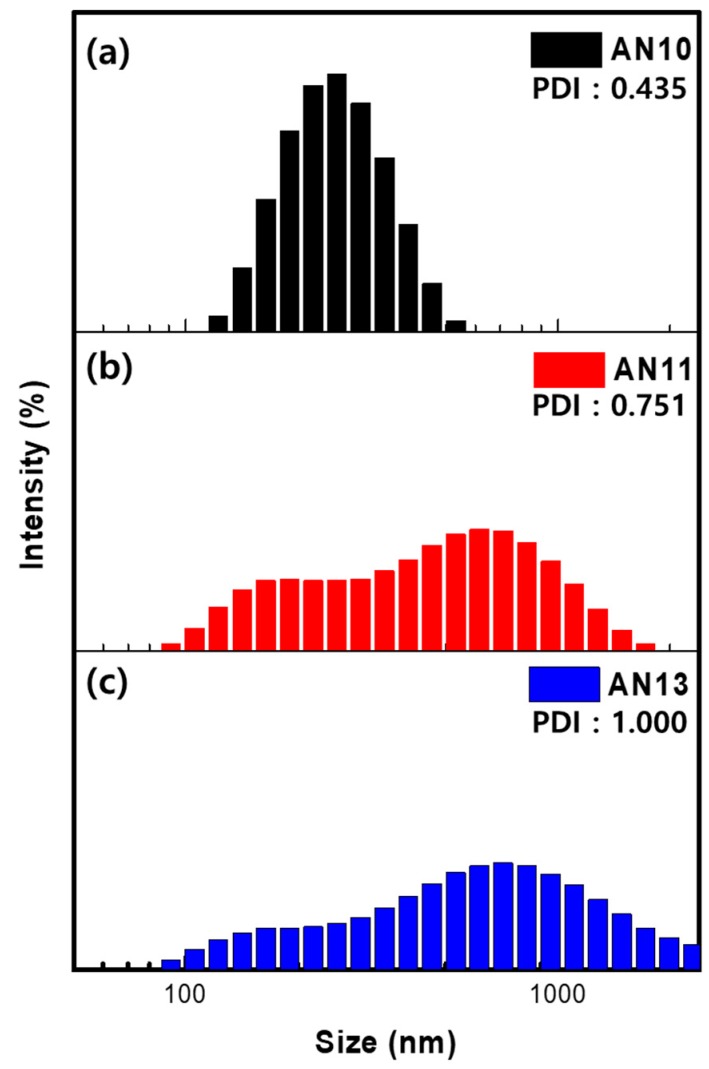
Zeta radius of samples with different ratios of acidic and neutral PEDOT:PSS. (**a**) AN10, (**b**) AN11, and (**c**) AN13 PEDOT:PSS.

**Figure 3 polymers-12-00129-f003:**
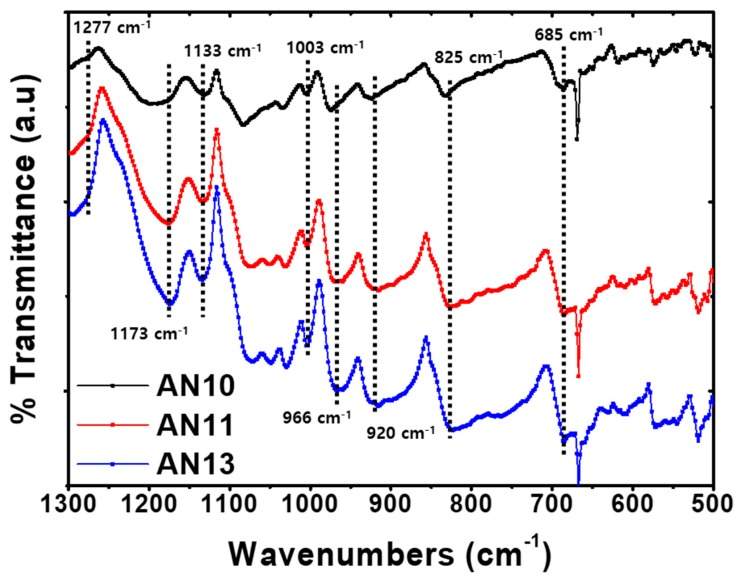
FTIR spectra of spin-coated AN10, AN11, and AN13 PEDOT:PSS films.

**Figure 4 polymers-12-00129-f004:**
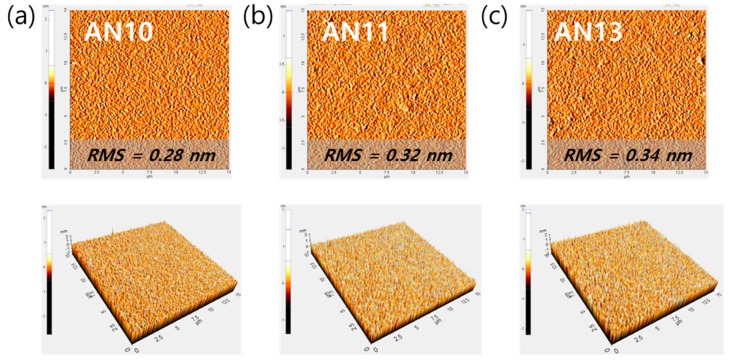
AFM topography (top) and three-dimensional views (bottom) of the surface of (**a**) AN10, (**b**) AN11, and (**c**) AN13 PEDOT:PSS.

**Figure 5 polymers-12-00129-f005:**
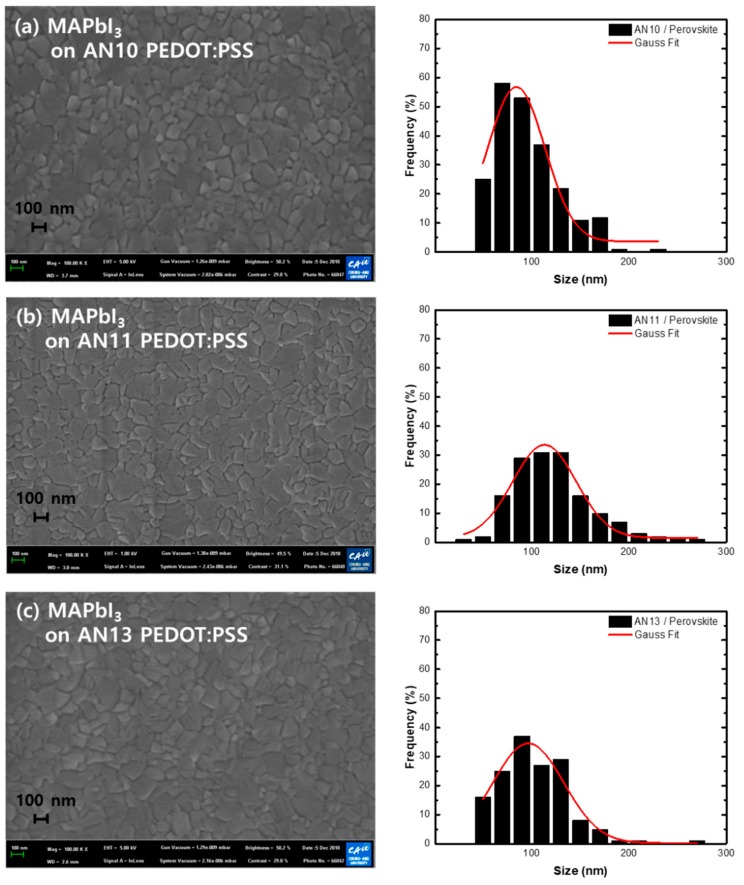
FE-SEM images of MAPbI_3_ crystals formed on thin films with different ratios of acidic and neutral PEDOT:PSS: (**a**) AN10, (**b**) AN11, and (**c**) AN13.

**Figure 6 polymers-12-00129-f006:**
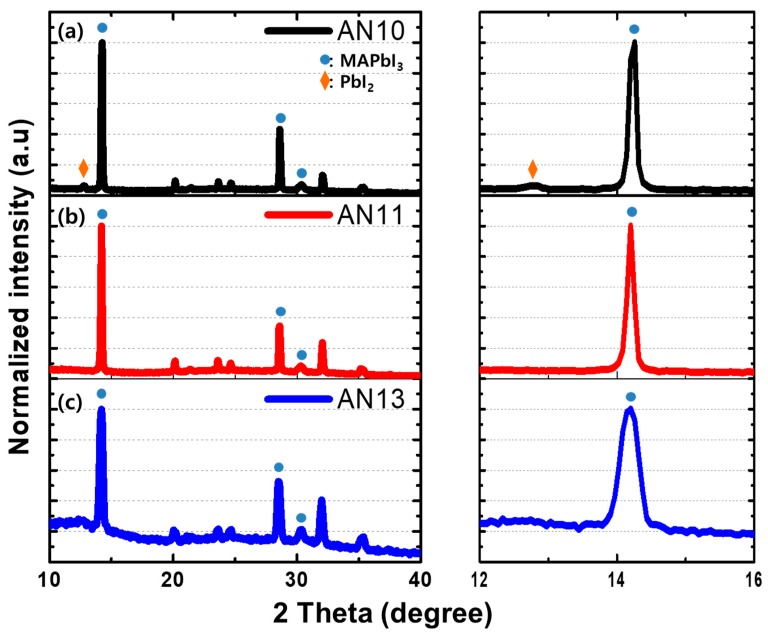
XRD patterns of the MAPbI_3_ perovskite films on various PEDOT:PSS layers: (**a**) AN10, (**b**) AN11, and (**c**) AN13.

**Figure 7 polymers-12-00129-f007:**
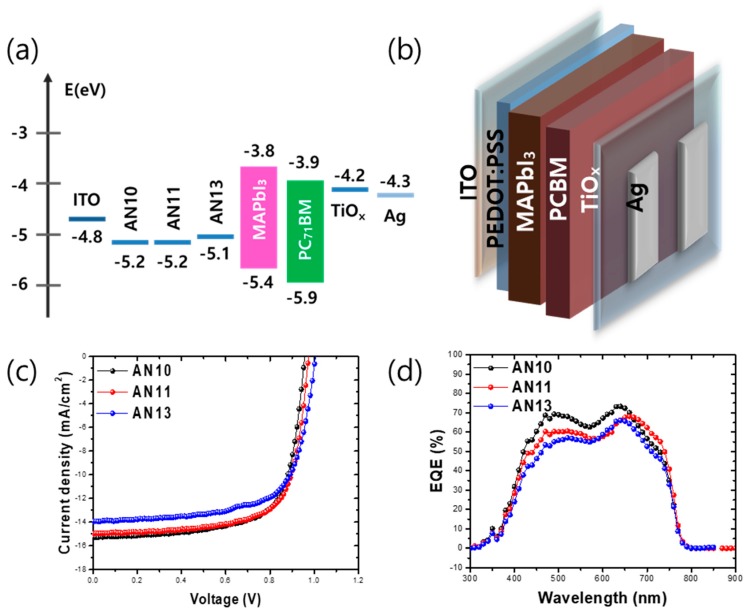
(**a**) Bandgap diagram of PSCs. (**b**) A schematic diagram of an inverted PSC. (**c**) Current-voltage (J–V) characteristics of PSCs under an AM 1.5 irradiation at 100 mW cm^−2^. (**d**) EQE (%) of PSCs depending on J–V characteristics.

**Figure 8 polymers-12-00129-f008:**
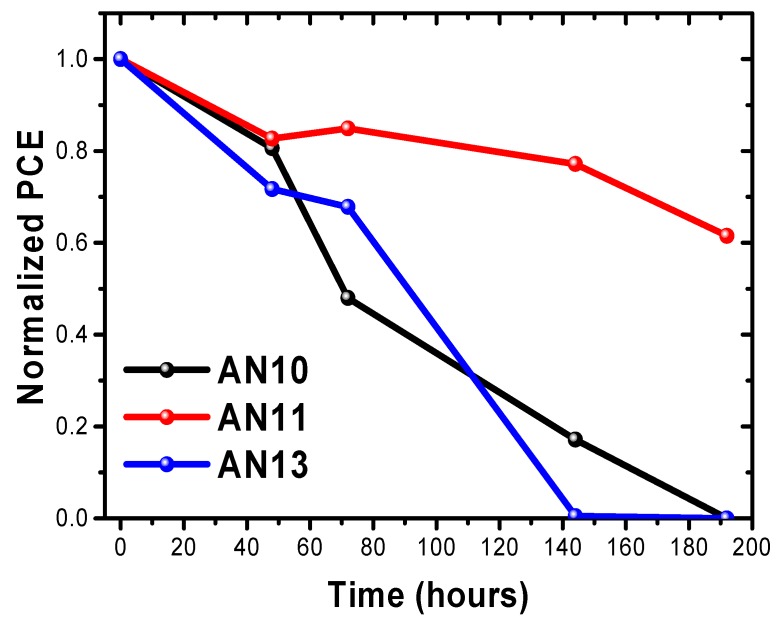
Long-term stability of AN10-, AN11-, and AN13-based PSCs.

**Table 1 polymers-12-00129-t001:** Electrical parameters of the AN10, AN11, and AN13 PEDOT:PSS-based PSCs.

Acidic: Neutral	V_OC_ (V)	Jsc (mA/cm^2^)	EQE (mA/cm^2^)	FF (%)	PCE (%)
AN10	0.957	15.28	14.56	70	10.31
AN11	0.975	14.98	13.69	71	10.34
AN13	1.006	13.94	12.59	68	9.60

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
