# Peer review of "Acidity Suppression of Hole Transport Layer via Solution Reaction of Neutral PEDOT:PSS for Stable Perovskite Photovoltaics"

_polymers, 2020, doi:10.3390/polym12010129_

Round 1
Reviewer 1 Report
Kim et al investigated the mixed neutral and acidic PEDOT:PSS and studied its influence on the performance of perovskite solar cell. They found that pH level of PEDOT:PSS affects the particle size of PEDOT:PSS colloidal and polymerization. The properties change in PEDOT:PSS film finally leads to morphology and crystallization of perovskite film, resulting in the enhanced performance of solar cell. The manuscript is well written and analyzed thoroughly to support the hypothesis and conclusion. Therefore, I recommend the manuscript to publish in “Polymers” with some minor changes.
PEDOT:PSS is also well know materials for thermoelectric generator. Hence, the author should add “thermoelectric generator” in line 41/42 and cite the related paper. For example: Macromolecular Materials and Engineering, 2018, 1700429, vol 303.
In line 186, the effect of grain boundaries on trap sites and charge transfer is also reported in the paper “Sol. RRL, 2019, 1900029, vol 3”. This paper is recommended to cite there.
Author Response
Dec 19, 2019
Manuscript Revision: Polymers
Dear Editor: Prof. Jeremy Li
From: Prof. Dong Hwan Wang
We have received a decision letter for our manuscript Ms. Ref. No.: polymers-666309, entitled “Acidity Suppression of Hole Transport Layer via Solution Reaction of Neutral PEDOT:PSS for Stable Perovskite Photovoltaics”. Based on our revision, we re-submitting the following manuscript to polymers by Minseong Kim, Minji Yi, Woongsik Jang, Jung Kyu Kim * and Dong Hwan Wang * from each affiliation.
From the reviewer’s constructive comments, we replied to all the responses in this cover letter and re-write in the main text (please see the red color). Therefore, we believe that our manuscript is worth to be published based on proper revision, and re-submit our fully revised manuscript to be considered in your journal. Finally, we look forward to hearing from your hopeful decision soon.
Thank you and best regards.
Sincerely Yours
Prof. Dong Hwan Wang
Distinguished Scholar, School of Integrative Engineering,
Chung-Ang University, 84 Heukseok-Ro,
Seoul, 156-756, Republic of Korea.
E-mail: king0401@cau.ac.kr
Research ID: http://www.researcherid.com/rid/J-1574-2014
We appreciate the referee’s comments and suggestions. After reading the comments from Reviewers carefully, all the authors feel that the questions raised can be properly answered with acceptable revision. We, therefore, submit the revised manuscript for publication in Polymers. All the changes made in the revised manuscript are in red fonts.
[Response to Reviewer #1]
Concerning the comment, “PEDOT:PSS is also well known materials for thermoelectric generator. Hence, the author should add “thermoelectric generator” in line 41/42 and cite the related paper. For example: Macromolecular Materials and Engineering, 2018, 1700429, vol 303.”
Our Response: As the reviewer’s comment, we added “thermoelectric generator” in line 42. And cited the related paper, “Macromolecular Materials and Engineering, 2018, 1700429, vol 303.” and “Chem. Mater. 2019, 31, 14, 5238-5244”.
(Please, refer to the follows page 1, line 39 in the revised paper.)
Concerning the comment, “In line 186, the effect of grain boundaries on trap sites and charge transfer is also reported in the paper “Sol. RRL, 2019, 1900029, vol 3”. This paper is recommended to cite there.”
Our Response: As a reviewer’s advice, we cited “Sol. RRL, 2019, 1900029, vol 3” in the revised paper.
(Please, refer to the follows page 6, line 28 in the revised paper.)
We thank the Referee for some constructive comments. We thank you for reconsidering the manuscript based upon our response to the Referee’s comments.

Reviewer 2 Report
This article reports on the acidity suppression of hole transport layer via solution reaction of neutral PEDOT: PSS for application in stable perovskite photovoltaic devices. The subject is very interesting because one can work out good quality reports on the application of polymeric materials in the field of photovoltaics, fuel cells and sensors etc. The article is well written and publishable in polymers after addressing the following issues.
Abstract
The authors have directly started abstract from PEDOT:PSS. It will be more appropriate to first write complete names and then synonames.
Introduction
Novelty of the work should be stressed with incorporation of more relevant literature.
Experimental
Please write complete name of the chemicals rather than simple formula Please provide information on the mode of FTIR spectra recorded. i.e Whether KBr pellets were used or ATR accessory.
Results and Discussion
In Figure 1 structure of PEDOT:PSS is presented. However, No description or method is provided in the whole manuscript about determination of this structure. The role of pH should be further discussed in the light of more relevant references. Figure 3. Presumably the FTIR spectra of PEDOT:PSS films were recorded on ITO substrate. However, no information is included on the presence/interaction of ITO bands. The authors write “To confirm the chemical stability of the PEDOT: PSS films, the polymerization degree of PEDOT was evaluated using the ratio of integration of the IR bands at 825 and 685 cm−1”. This statement is confusing and needs to be clarified. Also same PEDOT was used during preparation of all PEDOT: PSS films. How can the authors justify different degree of polymerization of PEDOT. It is not clear whether block copolymers or composites or simple mixture of PEDOT and PSS were deposited on the ITO surface.

Author Response
Dec 19, 2019
Manuscript Revision: Polymers
Dear Editor: Prof. Jeremy Li
From: Prof. Dong Hwan Wang
We have received a decision letter for our manuscript Ms. Ref. No.: polymers-666309, entitled “Acidity Suppression of Hole Transport Layer via Solution Reaction of Neutral PEDOT:PSS for Stable Perovskite Photovoltaics”. Based on our revision, we re-submitting the following manuscript to polymers by Minseong Kim, Minji Yi, Woongsik Jang, Jung Kyu Kim * and Dong Hwan Wang * from each affiliation.
From the reviewer’s constructive comments, we replied to all the responses in this cover letter and re-write in the main text (please see the red color). Therefore, we believe that our manuscript is worth to be published based on proper revision, and re-submit our fully revised manuscript to be considered in your journal. Finally, we look forward to hearing from your hopeful decision soon.
Thank you and best regards.
Sincerely Yours
Prof. Dong Hwan Wang
Distinguished Scholar, School of Integrative Engineering,
Chung-Ang University, 84 Heukseok-Ro,
Seoul, 156-756, Republic of Korea.
E-mail: king0401@cau.ac.kr
Research ID: http://www.researcherid.com/rid/J-1574-2014
We appreciate the referee’s comments and suggestions. After reading the comments from Reviewers carefully, all the authors feel that the questions raised can be properly answered with acceptable revision. We, therefore, submit the revised manuscript for publication in Polymers. All the changes made in the revised manuscript are in red fonts.
[Response to Reviewer #2]
Concerning the comment, “The authors have directly started abstract from PEDOT:PSS. It will be more appropriate to first write complete names and then synonames.”
Our Response: As the reviewer’s reasonable comment, we added the complete name of PEDOT:PSS at the start of the abstract.
(Please, refer to the follows page 1, line 12 in the revised paper.)
“Abstract: Poly(3,4-ethylenedioxythiophene):poly(4-styrenesulfonate) (PEDOT:PSS) is typically used hole transport layers (HTLs) which exhibits attractive mechanical, electrical properties and easy processability.”
Concerning the comment, “Novelty of the work should be stressed with incorporation of more relevant literature.”
Our Response: We stressed the novelty of the work with the more relevant literature in the revised manuscript as follows. The proton in acidic PEDOT:PSS affects the electrode and perovskite layer which affects the lifetime of device. And the inappropriate surface morphology of neutral PEDOT:PSS thin film induce low PCE in solar cells. The novelty of our approach is that the suppression of acidic properties and adjustment of the appropriate pH values in PEDOT:PSS for stable perovskite solar cells by a convenient solution reaction process. The optimized PEDOT:PSS layer affected the electrical properties of perovskite solar cell with enhanced stability.
The typo of the manuscript has been modified as follows.
(Please, refer to the follows page 2, line 11 in the revised paper.) “The residual acidic agents and proton in acidic PEDOT:PSS causes the problem of deterioration of ITO or top electrode and degradation of the perovskite which leads to decrease of perovskite crystal quality [27,32]. And the high viscosity of neutral PEDOT:PSS leads to unsuitable surface morphology for optoelectronic devices. We propose a convenient and successful approach to suppress the acidic property of PEDOT:PSS using a simple solution reaction that controls the volume ratio of two PEDOT:PSS solutions, acidic and neutral PEDOT:PSS.”
Concerning the comment, “Please write complete name of the chemicals rather than simple formula.”
Our Response: As the reviewer’s advice, we added the complete name of methylammonium iodide and lead iodide instead of CH3NH3I and PbI2.
(Please, refer to the follows page 2, line 25 in the revised paper.) “Methylammonium Iodide (CH3NH3I, Dyesol-Timo Co. Ltd., South Korea), Lead iodide (PbI2, 99%, Sigma Aldrich), γ-butyrolactone (GBL, Sigma Aldrich), and dimethyl sulfoxide (DMSO, Sigma Aldrich) were used to prepare the CH3NH3PbI3 (MAPbI3) precursor.” Concerning the comment, “Please provide information on the mode of FTIR spectra recorded. i.e Whether KBr pellets were used or ATR accessory.”
Our Response: The reviewer’s comment is reasonable. The FTIR spectra were obtained using NICOLET 6700 IR spectrometer and horizontal attenuated total reflectance (HATR) method of measurement was applied.
(Please, refer to the follows page 3, line 2 in the revised paper.)
“Fourier transform infrared (FTIR) spectra collected using horizontal attenuated total reflectance (HATR) were recorded on a NICOLET 6700 IR spectrometer.” Concerning the comment, “In Figure 1 structure of PEDOT:PSS is presented. However, No description or method is provided in the whole manuscript about determination of this structure.”
Our Response: The known PEDOT:PSS structure is further described and the explanation about gel-like particles in aqueous solution is also described. When PEDOT:PSS is deposited as thin film, PSS polymerize on the PEDOT with formation of π-stacking structure which shows hole transport property (Figure 1a). And when PEDOT:PSS is dispersed in aqueous solvents, PEDOT:PSS forms gel particle as shown in Figure 1b and the size distribution of gel particles is measured in Figure 2. We added the in the revised manuscript as follows. We added the description of the PEDOT:PSS structure in the revised manuscript as follows, and appreciate the reviewer’s comment.
(Please, refer to the follows page 3, line 13 in the revised paper.) “The PEDOT:PSS has the chemical structure and general arrangement [34-36] as shown in Figure 1a. The PEDOT:PSS consists of two ionic bond polymers presented in chemical structure. And when the PEDOT:PSS is deposited as a thin film, PSS polymerize onto the PEDOT which shows π-stacking structure which exhibit hole transport property [37,38].” (Please, refer to the follows page 4, line 4 in the revised paper.) “The PEDOT: PSS consists gel-like particles containing a PSS- shell which stabilizes PEDOT-rich particles in an aqueous solvent [43]. The size of the colloidal PEDOT:PSS gel particle in water is related to the ratio of PEDOT to PSS because increasing particle size increases the particle boundary surface [33,35,44] as depicted in Figure 1b.”
Concerning the comment, “The role of pH should be further discussed in the light of more relevant references.”
Our Response: As the reviewer’s comment, acidic property of PEDOT:PSS is further explained with more relevant references in the manuscript as follows. The residual acidic agents and proton in acidic PEDOT:PSS can potentially react with the bottom electrode, perovskite layer, and top electrode of perovskite solar cells (Nature Communications 2015, 6, 7348. Nature Communications 2015, 6, 7747. ACS Applied Materials & Interfaces 2018, 10, 18964-18973.). And it also results in the short lifetime of organic solar cell and light emitting diodes (Applied Physics Letters 2000, 77, 2255-2257. Energy & Environmental Science 2011, 4, 285-310. Advanced Materials 2002, 14, 206-209.). So the suppressing the acidity of PEDOT:PSS is important for stable photovoltaic devices.
(Please, refer to the follows page 3, line 16 in the revised paper.)
“The PEDOT:PSS exhibits strong acidic properties, and coating a solution with strong acidic PEDOT:PSS can deteriorate the ITO by penetrating the boundary between the solution and the thin film to promote ion release and impede charge flow [12,27,39-41]. Further, acidic property of PEDOT:PSS can cause the degradation of the top electrode or perovskite active layer and lead to inferior perovskite crystal quality[9,11,12,24,25,42].”
Concerning the comment, “Figure 3. Presumably the FTIR spectra of PEDOT:PSS films were recorded on ITO substrate. However, no information is included on the presence/interaction of ITO bands.”
Our Response: The authors appreciate the reviewer for logical prediction. We characterized the entire set of FTIR samples of PEDOT:PSS on silicon wafers were analyzed FTIR to estimate the structure and various bonds of PEDOT:PSS compounds. The silicon wafer has high transmittance in the IR range which is suitable for characterization of the deposited sample. However, the ITO substrate has an absorption range from about 800 to 1400 nm which makes the FTIR analysis difficult. By analyzing the PEDOT:PSS film on the silicon wafer, we could identify the functional groups of the PEDOT:PSS and PEDOT.
We added more explanation about the FTIR samples as follows.
(Please, refer to the follows page 5, line 7 in the revised paper.)
“To identify the degree of polymerization of PEDOT and the different vibrational modes of various bonds which are present in PEDOT:PSS layer, the spin-coated films on silicon wafer were studied using FTIR spectroscopy (Figure 3).” Concerning the comment, “The authors write “To confirm the chemical stability of the PEDOT: PSS films, the polymerization degree of PEDOT was evaluated using the ratio of integration of the IR bands at 825 and 685 cm−1”. This statement is confusing and needs to be clarified. Also same PEDOT was used during preparation of all PEDOT: PSS films. How can the authors justify different degree of polymerization of PEDOT.”
Our Response: The polymerization of PEDOT affects the charge transport and the efficient application in solar cell devices. As suggested, we clarified the sentence in the revised manuscript as follows.
We found that the polymerization degree of PEDOT can change by applying different solvent treatment methods or doping additive solvent with the same PEDOT:PSS (ACS Appl. Mater. Interfaces 2013, 5, 12811. Adv. Electron. Mater., 1: 1500017. Polymers, 11(6), 1034). In this research, we used different PEDOT:PSS (AI 4083 and neutral PEDOT:PSS) which has different properties. These different properties could affect the polymerization of PEDOT and we could calculate the different degrees of polymerization. Also, the different degrees of polymerization can be justified by calculating the ratio of integration of the IR bands at 825 and 685 cm−1 (Nanoscale research letters, 9(1), 557. Synthetic metals, 100(3), 285-289.).
(Please, refer to the follows page 5, line 18 in the revised paper.) “To compare electrochemical polymerization of each PEDOT:PSS films, the polymerization degree of PEDOT was evaluated using the ratio of integration of the IR bands at 825 and 685 cm−1; a lower intensity ratio indicates a higher formation of polymerization reaction in PEDOT [50-52].” (Please, refer to the follows page 5, line 26 in the revised paper.) “Our results indicate that the varied properties of AN10, AN11, and AN13 induce the different degree of polymerization which affects the carrier mobility of PEDOT:PSS [51,53,54]. Based on the above result, the AN11 PEDOT:PSS is expected to exhibit enhanced charge carrier mobility because of improved conjugation and polymerization with π-π stacking structure which improves the electrical properties of PSCs.” Concerning the comment, “It is not clear whether block copolymers or composites or simple mixture of PEDOT and PSS were deposited on the ITO surface.”
Our Response: The PEDOT:PSS dispersion is described as composition of gel-like particles with hydrophobic PEDOT-rich core and a hydrophilic PSS-rich shell. These composites of gel-particle deposit as a pancake-like morphology, which imply randomly distributed in the dispersion (Adv. Funct. Mater. 19, 1215–1220 (2009). Adv. Mater. 19, 1196–1200 (2007). J. Phys. Chem. B 108, 18820–18825 (2004)). After the thermal annealing and additional polymerization, the phase separation of PEDOT and PSS occurs which induces additional polymerization of PEDOT. Furthermore, it increases the continuous π-stacking of PEDOT:PSS and provides a pathway where charge carrier can flow.
(Please, refer to the follows page 5, line 1 in the revised paper.)
“As the gel particle in aqueous solution is deposited and annealed forming thin PEDOT:PSS film, the PEDOT chains interact with the neighboring PEDOT-rich domains and bring the conducting domains closer by further polymerization [44,47,48]. And the thermal annealing evaporates PSS-rich domains which improves connectivity and charge transport of conducting domains [47]. After the annealing and additional polymerization, the phase separation of PEDOT and PSS occurs. Furthermore, it increases the continuous π-stacking of PEDOT:PSS and provides a pathway where charge carrier can flow [37,38,49].”
We thank the Referee for some constructive comments. We thank you for reconsidering the manuscript based upon our response to the Referee’s comments.

Round 2
Reviewer 2 Report
The authors have tried to revise the manuscript in the light of my previous comments. However, some comments are not addressed. For example the authors write in the revised manuscript that
When the PEDOT:PSS is deposited as a thin film, PSS polymerize onto the PEDOT which shows π-stacking structure which exhibit hole transport property. How does PSS polymerize onto the PEDOT provided that PSS is a polymer by itself? the PEDOT chains interact with the neighboring PEDOT-rich domains and bring the conducting domains closer by further polymerization. Is it really possible to further polymerize a polymer without presence of monomers? To compare electrochemical polymerization of the each PEDOT:PSS films, the polymerization degree of PEDOT was evaluated using the ratio of integration of the IR bands at 825 and 685 cm−1. I believe that no electrochemical polymerization has been carried out in this study. How do the authors compare electrochemical polymerization of the each PEDOT:PSS films? Still it is not clear whether block copolymers or composites or simple mixture of PEDOT and PSS were deposited on the ITO surface.

Author Response
We appreciate the referee’s comments and suggestions. After reading the comments from Reviewers carefully, all the authors feel that the questions raised can be properly answered with acceptable revision. We, therefore, submit the revised manuscript for publication in Polymers. All the changes made in the revised manuscript are in red fonts.
[Response to Reviewer #2]
Concerning the comment, “When the PEDOT:PSS is deposited as a thin film, PSS polymerize onto the PEDOT which shows π-stacking structure which exhibit hole transport property. How does PSS polymerize onto the PEDOT provided that PSS is a polymer by itself.”
Our Response: Based on the reviewer’s question, we modified ambiguous words. The PEDOT:PSS is composed of poly(3,4-ethylenedioxythiophene) (PEDOT) polymerized with poly(4-styrenesulfonate) (PSS) [ACS Applied Materials & Interfaces 2011, 3, 43-49. Organic Electronics 2006, 7, 387-396]. While the PEDOT:PSS film is deposited as a thin film, π-stacking structure between PEDOT and PSS is formed because the PEDOT is bonded to PSS chains via Coulomb interactions. We clarified the sentence about the interaction between PEDOT and PSS chain in the revised manuscript as follows.
(Please, refer to the follows page 3, line 13 in the revised paper.)
“And when the PEDOT:PSS is deposited as a thin film, since the PEDOT is bonded to PSS chains via Coulomb interactions, π-stacking structure between PEDOT and PSS is induced, which improves hole transport property and conductivity. [37,38].” Concerning the comment, “the PEDOT chains interact with the neighboring PEDOT-rich domains and bring the conducting domains closer by further polymerization. Is it really possible to further polymerize a polymer without presence of monomers?”
Our Response: The reviewer’s comment is reasonable. It is hard to polymerize each polymer without the presence of monomers. We intended to explain the increase of size in PEDOT-rich grains by coalescence of PEDOT:PSS particles formation of thin film of PEDOT:PSS [Reynolds, J. R., Conjugated Polymers: Properties, Processing, and Applications; 2019. Organic Electronics 2009, 10, 61-66]. As the PEDOT:PSS is deposited and annealed, PSS is slightly evaporated by thermal annealing in the PEDOT:PSS film. This led to the coalescence of PEDOT:PSS particles by softening the PSS and PEDOT particles, which increase the connectivity of the conducting domains [Advanced Science 2019, 6, 1900813 Organic Electronics 2009, 10, 61-66. Journal of Materials Science 2013, 48, 3528-3534. Synthetic Metals 2003, 139, 569-572].
(Please, refer to the follows page 4, line 20 in the revised paper.)
“As the gel particle polymer composite in aqueous solution is deposited and annealed forming thin PEDOT:PSS film [44,47-50], the PEDOT chains interact with the neighboring PEDOT-rich domains and bring the conducting domains closer by coalescence of PEDOT:PSS particles. And the thermal annealing evaporates and softens PSS-rich domains which improve connectivity and charge transport of conducting domains [48,51,52]. Concerning the comment, “To compare electrochemical polymerization of the each PEDOT:PSS films, the polymerization degree of PEDOT was evaluated using the ratio of integration of the IR bands at 825 and 685 cm-1. I believe that no electrochemical polymerization has been carried out in this study. How do the authors compare electrochemical polymerization of the each PEDOT:PSS films?”
Our Response: As the reviewer’s constructive comments, we added more explanation about the degree of polymerization. In PEDOT, electrochemical polymerization occurs at the α,α′-positions in polythiophene [Synthetic Metals 1999, 100, 285-289]. And the band at 685 and 825 cm−1 indicate the characteristic bands of stretching vibrations of the C-S-C bond. The average degree of polymerization of PEDOT can be evaluated using the equation as follows [Polymer Chemistry 2012, 3, 436-449. Nanoscale Research Letters 2014, 9, 557. Synthetic Metals 2005, 155, 232-239]:
Degree of polymerization
where R is the integrated intensity ratio of the IR bands at 685 and 825 cm−1, and R0 is the value of R determined for α-sexithiophene, 1.07. By the above equation, the degree of polymerization can be calculated. The higher degree of polymerization resulted from a relatively lower ratio of integrated intensity at 685 and 825 cm−1. We could compare the degree of polymerization of PEDOT by the above method and added the revised manuscript as follows.
(Please, refer to the follows page 5, line 15 in the revised paper.) “To compare the degree of electrochemical polymerization occurs at the α,α’-positions in polythiophene [54], the polymerization degree of PEDOT was evaluated from the ratio of integration of the infrared bands [55]. The average degree of polymerization of PEDOT can be evaluated using the equation as follows [56-58]:
Degree of polymerization
where R is the integrated intensity ratio of the IR bands at 685 and 825 cm−1 which indicates the characteristic bands of stretching vibrations of the C-S-C bond and R0 is 1.07 which is determined for α-sexithiophene.”
Concerning the comment, “Still it is not clear whether block copolymers or composites or simple mixture of PEDOT and PSS were deposited on the ITO surface.”
Our Response: The PEDOT:PSS is polymer composite, which consists of polyanion rich shell with the PEDOT enriched core [ACS Applied Materials & Interfaces 2012, 4, 2551-2560. Advanced Materials 2007, 19, 1196-1200, Advanced Functional Materials 2009, 19, 1215-1220]. These composites show the structure of gel particle in aqueous solution. As mentioned above, the PEDOT:PSS film is composed of PEDOT grains and PSS chains, so, the polymer grains are defined by the PSS random coil with PEDOT chains ionically attached along with them [Synthetic Metals 2003, 139, 1-10]. The gel particle composite of PEDOT:PSS is depicted and modified in Figure 1.
(Please, refer to the follows page 4, line 20 in the revised paper.) “As the gel particle polymer composite in aqueous solution is deposited and annealed forming thin PEDOT:PSS film [44,47-50], the PEDOT chains interact with the neighboring PEDOT-rich domains and bring the conducting domains closer by coalescence of PEDOT:PSS particles.”
Figure 1. (a) Structure and schematic representation of PEDOT:PSS. (b) The solution fabrication process and polymer composite structure of pH-controlled PEDOT:PSS for inverted PSCs.
We thank the Referee for some constructive comments. We thank you for reconsidering the manuscript based upon our response to the Referee’s comments.

Round 3
Reviewer 2 Report
The manuscript has been revised extensively and is acceptable for publication in Polymers